# Blood pressure changes during the first 24 hours of life and the association with the persistence of a patent ductus arteriosus and occurrence of intraventricular haemorrhage

**Robert Boldt**[1]*, **Pauliina M. Mäkelä**[1], **Lotta Immeli**[1], **Reijo Sund**[2], **Markus Leskinen**[1], **Päivi Luukkainen**[1], **Sture Andersson**[1]

**1** New Children's Hospital, Pediatric Research Center, University of Helsinki, and Helsinki University Hospital, Helsinki, Finland, **2** University of Eastern Finland, School of Medicine, Kuopio, Finland

* Robert.Boldt@hus.fi

## Abstract

Very low birthweight (VLBW) infants are at risk of intraventricular haemorrhage (IVH) and delayed closure of ductus arteriosus. We investigated mean arterially recorded blood pressure (MAP) changes during the first day of life in VLBW infants as potential risk factors for a patent ductus arteriosus (PDA) and IVH. This retrospective cohort study exploring MAP changes during adaption and risk factors for a PDA and IVH comprised 844 VLBW infants admitted to the Helsinki University Children's Hospital during 2005–2013. For each infant, we investigated 600 time-points of MAP recorded 4–24 hours after birth. Based on blood pressure patterns revealed by a data-driven method, we divided the infants into two groups. Group 1 (n = 327, mean birthweight = 1019 g, mean gestational age = 28 + $^{1/7}$ weeks) consisted of infants whose mean MAP was lower at 18–24 hours than at 4–10 hours after birth. Group 2 (n = 517, mean birthweight = 1070 g, mean gestational age = 28 + $^{5/7}$ weeks) included infants with a higher mean MAP at 18–24 hours than at 4–10 hours after birth. We used the group assignments, MAP, gestational age at birth, relative size for gestational age, surfactant administration, inotrope usage, invasive ventilation, presence of respiratory distress syndrome or sepsis, fluid intake, and administration of antenatal steroids to predict the occurrence of IVH and use of pharmacological or surgical therapy for a PDA before 42 weeks of gestational age. Infants whose mean MAP is lower at 18–24 hours than at 4–10 hours after birth are more likely to undergo surgical ligation of a PDA (odds ratio = 2.1; CI 1.14–3.89; p = 0.018) and to suffer from IVH (odds ratio = 1.83; CI 1.23–2.72; p = 0.003).

## Introduction

A delayed patent ductus arteriosus (PDA) closure [1, 2] and intraventricular haemorrhage (IVH) [3] are more common in preterm than in term infants during the neonatal period. Respiratory distress syndrome (RDS), sepsis, and low birthweight further increase the risk of a

**Data Availability Statement:** All relevant data are within the paper and its Supporting Information files.

**Funding:** This study was funded by Lastentautien tutkimussäätiö (Foundation for Pediatric Research, https://www.lastentautientutkimussaatio.fi), Finska Läkaresällskapet, and a Special Governmental Subsidy for Clinical Research, through grants awarded to RB. The funders had no role in study design, data collection and analysis, decision to publish, or preparation of the manuscript.

**Competing interests:** The authors have declared that no competing interests exist.

PDA [2, 4–6] and IVH [3]. On the contrary, restricting fluid intake during the first days of life may decrease the risk of developing a PDA [7, 8]. A PDA is associated with bronchopulmonary dysplasia, necrotising enterocolitis, and grade III–IV IVH [6, 9].

The first hours in the life of a very low birthweight infant (birthweight less than 1500 g) offer a unique opportunity to study adaption to extrauterine life. For instance, we can closely monitor blood pressure changes during adaption as invasive blood pressure measurement is used in most very low birthweight (VLBW) infants. In extremely preterm infants, arterial blood pressure gradually increases during the first 24 hours of life [10]. However, the mean arterial blood pressure (MAP) seems to reach a lower level in infants with a PDA [11]. A MAP of less than 33 mmHg at 13 to 15 hours of life for infants with a gestational age of less than 32 weeks identifies infants at risk of a significant PDA [12, 13]. Additionally, infants with hypotension during the first 24 hours have a higher incidence of intraventricular haemorrhage, BPD, and death [14]. Investigating more complex blood pressure changes might be beneficial when predicting the need for a PDA intervention or risk for IVH. Data-driven methods offer an effective way to organise complex data, to find hidden patterns, and for instance, to explore changes in blood pressure.

The neonatal intensive care unit (NICU) of the Helsinki University Children's Hospital has collected a broad set of clinical parameters for all infants with a birthweight of less than 1500 g [15]. This study hypothesised that a data-driven method could reveal hidden patterns in arterially recorded blood pressure during the first 4–24 hours of life in 844 VLBW infants. Furthermore, we investigated whether these blood pressure patterns have clinical relevance for predicting PDA interventions and the occurrence of IVH.

## Methods

We conducted this study as part of the Big Data–Tiny Infants research project, including infants with a birthweight of less than 1500 g [16, 17]. We retrieved our data from two databases: 1) The Finnish Medical Birth Register data on premature infants, managed by the Finnish Institute for Health and Welfare, Finland, and 2) The Helsinki University Children's Hospital's NICU electronic patient information system, Centricity Critical Care Clinisoft (GE Healthcare, Chicago, IL). The Register authority and the Ethics Committee of the Helsinki University Hospital approved this study. As all the data were collected retrospectively and pseudonymised, informed consent was waived.

### Data retrieved from the Finnish Medical Birth Register data on premature infants

The Finnish Medical Birth Register data on premature infants contains prenatal and neonatal information on infants born in Finland with a birthweight of less than 1501 g or a gestational age at birth of less than 32 + $^{0/7}$ weeks [18]. We retrieved gestational age at birth, birthweight, exposure to antenatal steroids, whether the infants received surfactant or invasive ventilation, and whether the infants suffered from RDS from the Finnish Medical Birth Register data on premature infants. Furthermore, we obtained information on when and if the infant had undergone surgical ligation to achieve ductal closure and the occurrence and grade of IVH before 42 weeks of gestational age. We retrieved data on pharmacological PDA interventions from the Helsinki University Children's Hospital's NICU electronic patient information system.

Duration of gestation was determined from the first day of the last menstrual period. In 87%, this was confirmed or adjusted if needed by an ultrasound examination during the first trimester. We defined small for gestational age (SGA) as a birthweight Z-score of less than −2 standard deviations on the Finnish growth charts [19]. RDS was defined as the presence of

clinical symptoms, typical chest x-ray findings, and laboratory abnormalities from an impaired gas exchange [20]. IVH was graded, according to Papile [21].

## Data retrieved from the Helsinki University Children's Hospital's NICU electronic patient information system

The Helsinki University Children's Hospital's NICU electronic patient information system stores patient monitoring data, observation variables, and laboratory results collected during a NICU stay. The electronic patient information system also contains computerised medication administration records with detailed information of all preparations given to each infant: name of preparation, the amount, and administration time. Monitor data, such as heart rate, airway pressure, oxygen saturation, temperature, and arterially measured blood pressure, were automatically recorded and stored as 2-minute averages of median values for 10-second intervals.

From the electronic patient information system, we obtained for each VLBW infant included in this study cohort the following information: Mean airway pressure for infants with invasive ventilation and fluid intake during the first 24 hours of life, whether infants had vasoactive support during the first 24 hours after birth, drug of choice and administration time for a PDA intervention, positive blood cultures, and laboratory results suggestive of sepsis.

Lastly, we retrieved 600 measurements of systolic, mean, and diastolic arterial blood pressure for each infant, representing blood pressure changes 4–24 hours after birth. We chose the 4–24 hours data range to optimise the trade-offs between the number of available infants and time-points of blood pressure measurements.

For each VLBW infant, we calculated the total amount of enteral and intravenous fluids received from nutrition, medicines, and other sources [16]. According to our NICUs practice, we considered preparations administered intravenously to consist entirely of fluids. However, we estimated the water content of milk and formulae as 80% [16].

The definition of sepsis, containing both early and late sepsis, was a positive blood culture and clinical symptoms [22] before 90 days of age or before a surgical PDA ligation, whichever came first. We excluded culture-negative results but were unable to exclude the uncertain blood cultures from the culture-positive results. Thus, we demanded that the C-reactive protein counts were elevated (more than 5 mg/l) during the preceding or succeeding two days relative to the positive blood culture to exclude false positives. In sepsis, elevated C-reactive protein (more than 5 mg/l) levels offer a high specificity [23], and the negative predictive value of serial C-reactive protein measurements could be as high as 99% [24].

The primary vasoactive agent in our NICU is dopamine. Additional vasoactive agents are noradrenalin, adrenalin and dobutamine. The MAP threshold for initiating vasoactive support in our NICU typically equals the infant's gestational age in weeks.

According to our NICUs practice, diagnosis of a significant PDA requires indications of left heart volume overload, pulmonary hyperperfusion and focused cardiac ultrasound showing a moderate or large PDA. Surgical intervention is indicated if a significant PDA persists despite pharmacological interventions or if pharmacological interventions are contraindicated.

## Study cohort and exclusion criteria

During years 2005–2013, 1227 infants with a birthweight of less than 1500 g were admitted to the NICU of Helsinki University Children's Hospital. The exclusion criteria for the study cohort are depicted in Fig 1. After excluding infants not registered to either the Finnish Medical Birth Register data on premature infants or the NICU electronic patient information system, infants with congenital malformations or chromosomal anomalies, infants deceased before 24 hours of age, infants with a pharmacological PDA intervention before 24 hours of

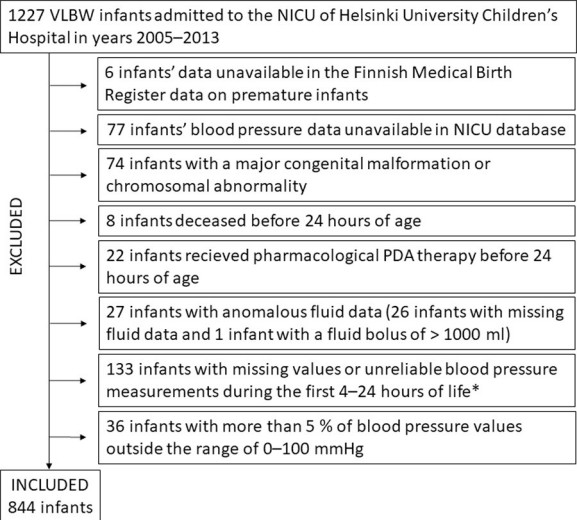

**Fig 1. Flowchart of the exclusion criteria for the study cohort.** *More than 5% of mean arterial blood pressure values missing or a constant value for more than 1/3 of the mean arterial blood pressure measurements.

age, infants with incomplete fluid data, and infants with insufficient or unreliable blood pressure data, 844 infants remained in the study cohort. In addition to the eight infants discarded from the study as they deceased before 24 hours of age, 61 infants died before 42 weeks of gestational age. Of these 61 infants, 29 died during the first week, 14 during the second week, three during the third week, three during the fourth week and 12 after the fourth week.

## K-means clustering

For grouping purposes, we aimed at finding hidden blood pressure patterns with a data-driven method, k-means clustering. K-means clustering is one of the most popular data mining algorithms used to classify clinical data [25, 26]. The k-means clustering allocated infants with similar blood pressure patterns to the same clusters. In contrast, infants assigned to separate clusters had distinct blood pressure patterns. Preprocessing of the blood pressure measurements included smoothing the data (20-minute Gaussian-weighted moving average filter). Using the MAP values, we grouped the infants with k-means clustering. For the k-means clustering, our distance measure was a time-domain correlation in blood pressure trends. We used the silhouette method to decide the number of clusters [27].

## Logistic regression

We fitted a multinomial logistic regression model to find the variables associated with pharmacological and surgical PDA interventions before 42 weeks of gestational age. Needing both pharmacological and surgical treatment indicated failed pharmacological treatment followed by surgical treatment. The multinominal model was fitted for the following comparison: no treatment versus pharmacological treatment; pharmacological treatment versus pharmacological and surgical treatment [28]. The multinominal model could not account for the infants who underwent direct surgical ligation of a PDA. Thus, we fitted a separate logistic regression model for no treatment versus a surgical PDA treatment, including both infants who underwent direct surgical ligation and surgical ligation after pharmacological therapy (S1 Table). We also fitted a separate logistic regression model for no IVH vs IVH of any grade.

Additionally, we fitted a multinominal logistic regression model for no IVH vs IVH of any grade and IVH of grades I-II vs IVH of grades III-IV (S2 Table).

In all logistic regression models, we used the following predictor variables; 1) gestational age at birth (less than 28 weeks or 28 + 0/7 weeks and more), 2) whether the infant was appropriate for gestational age or SGA, whether the infant suffered from 3) RDS or 4) sepsis before 90 days of age or before a surgical PDA ligation, 5) whether the mother received antenatal steroid treatment, 6) whether the infant received more than 120 ml/kg of fluids during the first 24 hours of life, 7) infants' group assignment according to the blood pressure patterns, 8) whether the mean MAP during the 13 to 15th hour after birth was either lower or equal to or higher than 33 mmHg, 9) whether the infant received pharmacological vasoactive support, 10) whether the infant received surfactant, and 11) whether the infant was invasively ventilated.

Based on the gestational age, a mean blood pressure of less than 33 mmHg at 13 to 15 hours of life offers the best diagnostic accuracy for early prediction of a significant PDA [12]. We used the upper limit (120 ml/kg) suggested by the ESPGHAN/ESPEN/ESPR guidelines to indicate excessive fluid administration during the first 24 hours of life [29].

# Results

## Cluster analysis

The silhouette method indicated that the most suitable number of clusters for the k-mean clustering analysis was two. The k-means clustering method successfully grouped infants with similar blood pressure time series. In contrast, infants assigned to separate clusters had dissimilar blood pressure time series (see correlation matrix in Fig 2B). Cluster one contained 335 infants who, after an initial rise, had a declining blood pressure trend throughout the study period (Fig 2C). Cluster two contained 509 infants who displayed a steadily rising blood pressure trend throughout the study (Fig 2D). The clustering was stable, as shown by repeating the clustering analysis with 4000 bootstrap samples (S1 Fig).

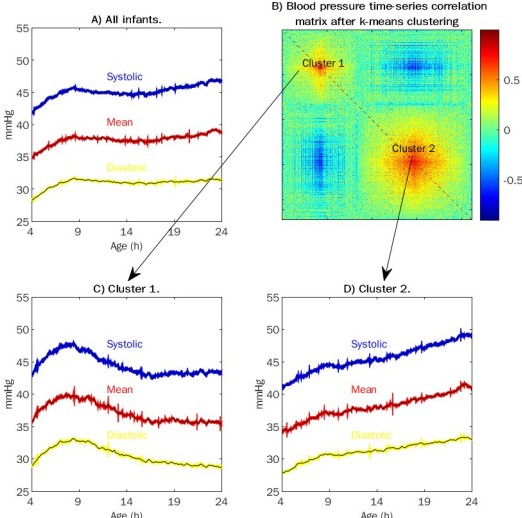

**Fig 2. Blood pressure patterns during the first 4–24 hours after birth revealed by k-means clustering.** (A) The mean and the standard error for the systolic (blue), mean (red), and diastolic (yellow) arterially recorded blood pressure during the first 4–24 hours after the birth of all the 844 very low birthweight infants in the study cohort. (B) The correlation matrix shows how k-means clustering divided the 844 infants into two clusters based on the infants' blood pressure time series. The size of the correlation matrix corresponds to the number of infants. (C) Arterially recorded blood pressure for infants in cluster 1 (n = 335). (D) Arterially recorded blood pressure for infants in cluster 2 (n = 509).

The blood pressure level was closely related to gestational age; only two infants with a gestational age less than 27 weeks had a mean MAP higher than the mean MAP for all infants with a gestational age over 31 weeks. Despite the age-dependent blood pressure level, when repeating the clustering across different gestational ages, the trend of one cluster of infants with a declining and one cluster of infants with a steadily rising blood pressure time series remained (S2 Fig).

The blood pressure of infants in cluster 1 peaked approximately seven hours after birth. In fact, we could classify 95% of the infants into the two clusters by comparing the mean MAP at 4–10 hours to the mean MAP at 18–24 hours after birth. To facilitate our results' reproducibility and clinical applicability of similar blood pressure data in other settings, we defined Group 1 (n = 327) of the infants whose mean MAP at 18–24 hours after birth was lower than at 4–10 hours after birth. Group 2 (n = 517) consisted of all other infants, i.e., the infants whose mean MAP at 18–24 hours after birth was higher than or the same as at 4–10 hours after birth.

Sampling every fifth blood pressure value (one value for every 10 minutes) still classified 95% of the infants into the two clusters. Sampling every fifteenth blood pressure value (one value for every 30 minutes) classified 93% of the infants into the two clusters. Sampling every thirtieth blood pressure value (one value for each hour) classified 88% of the infants into the two clusters, while sampling one value for every four hours classified 78% of the infants into the two clusters. Table 1 displays the clinical characteristics of the study groups.

## Factors associated with the treatment of a PDA and factors associated with the presence of IVH

The significant explanatory variables associated with pharmacological treatment of a PDA were gestational age at birth of less than 28 weeks, presence of RDS, and whether infants required inotropes or surfactant after birth.

After a failed pharmacological PDA treatment, the significant explanatory variables associated with surgical ligation were gestational age at birth of fewer than 28 weeks, fluid administration of more than 120 ml/kg and group assignment. Infants in Group 1 (infants with lower mean MAP at 18–24 hours than at 4–10 hours after birth) had an odds ratio (OR) of 2.1 to need surgical treatment after pharmacological treatment of a PDA.

To achieve pharmacological ductal closure, most infants (n = 210) received indomethacin (dose for infants with a weight of less than 1000 g 0.1 mg/kg at 0, 8 and 20 hours: initial dose for infants with a weight of more than 1000 g 0.2 mg/kg, supplementary doses 0.1 mg/kg at 12 and 24 hours). Following a change in our NICU's policy, 55 infants received, between May 2008 and March 2010, ibuprofen (initial dose 10 mg/kg, supplementary doses 5 mg/kg at 24 and 48 hours) to achieve ductal closure.

Only five infants had a positive blood culture before pharmacological PDA interventions. Overall, there were only 13 early-onset (positive blood culture before 72 hours of age) sepsis cases in the study population. Table 2 depicts the multinominal logistic regression analysis results for variables predicting interventions of a PDA.

Table 3 depicts the multinominal logistic regression analysis results for variables predicting the presence of IVH. The significant factors associated with IVH were the gestational age at birth of less than 28 weeks, presence of RDS, fluid intake of more than 120 ml/kg during the first day of life, and group assignment. Belonging to Group 1 (infants with lower mean MAP at 18–24 hours than at 4–10 hours after birth) increased the likelihood of having IVH. The multinominal logistic regression analysis results for variables predicting the risk of having no intraventricular haemorrhage (IVH) or grade I–II IVH vs grade III–IV IVH is included in the S2 Table.

We did a post hoc receiving operating characteristics (ROC) analysis as a test of model adequacy of the logistic regression models (Fig 3) [30]. The area under the ROC curve for

**Table 1. Clinical characteristics of the study groups.**

| | All | MAP 18–24 < MAP 4–10 hours after birth (Group 1) | MAP 18–24 ≥ MAP 4–10 hours after birth (Group 2) | Difference/OR (CI), p-value (*two-sample t-test/†Fischer's exact test) |
|---|---|---|---|---|
| Number of infants | 844 | 327 | 517 | |
| Mean gestational age at birth and range (weeks $+^{days}$) | $28 + ^{3/7}$ | $28 + ^{1/7}$ ($23 + ^{0/7}$–$34 + ^{6/7}$) | $28 + ^{5/7}$ ($23 + ^{0/7}$–$34 + ^{6/7}$) | -4.49 days (-6.92– -2.06), $p < 0.001$* |
| Mean birthweight and range (g) | 1045 | 1019 (375–1495) | 1070 (430–1495) | -51 g (-88.93– -13.06), $p = 0.008$* |
| Only pharmacological PDA treatment, n (%) | 182 (22%) | 65 (20%) | 117 (23%) | OR = 0.85 (0.6–1.19), $p = 0.39$† |
| Only surgical PDA treatment, n (%) | 56 (7%) | 30 (9%) | 26 (5%) | OR = 1.91 (1.11–3.29), $p = 0.023$† |
| Both pharmacological and surgical PDA treatment, n (%) | 83 (10%) | 44 (13%) | 39 (8%) | OR = 1.91 (1.21–3), $p = 0.006$† |
| Respiratory distress syndrome, n (%) | 420 (50%) | 175 (54%) | 245 (47%) | OR = 1.28 (0.97–1.69), $p = 0.09$† |
| Sepsis, n (%) | 149 (18%) | 51 (16%) | 98 (19%) | OR = 0.79 (0.55–1.14), $p = 0.229$† |
| Small for gestational age, n (%) | 216 (26%) | 74 (23%) | 142 (27%) | OR = 0.77 (0.56–1.07), $p = 0.124$† |
| Received antenatal corticosteroids before birth, n (%) | 808 (96%) | 308 (94%) | 500 (97%) | OR = 0.55 (0.28–1.08), $p = 0.083$† |
| Fluid > 120 ml/kg during first 24 hours, n (%) | 414 (49%) | 196 (60%) | 218 (42%) | OR = 2.05 (1.55–2.72), $p < 0.001$† |
| Time from birth to the first dose of pharmacological PDA treatment, mean and range (days) | 2.9 | 2.9 (1–11.2) | 2.9 (1–15.9) | -0.05 days (-0.57–0.47), $p = 0.852$* |
| Time from birth to surgical PDA ligation, mean and range (days) | 10.4 | 9.4 (1.2–42.4) | 11.4 (1.4–39.5) | -1.93 days (-4.87–1.01), $p = 0.196$* |
| MAP < 33 mmHg 13–16 hours after birth, n (%) | 163 (19%) | 82 (25%) | 81 (16%) | OR = 1.8 (1.28–2.54), $p = 0.001$† |
| Inotrope during first 24 hours, n (%) | 418 (50%) | 195 (60%) | 223 (43%) | OR = 1.95 (1.47–2.58), $p < 0.001$† |
| IVH grades I–II, n (%) | 74 (9%) | 38 (12%) | 36 (7%) | OR = 1.76 (1.09–2.84), $p = 0.024$† |
| IVH grades III–IV, n (%) | 68 (8%) | 40 (12%) | 28 (5%) | OR = 2.43 (1.47–4.03), $p = 0.001$† |
| Invasive ventilation, n (%) | 523 (62%) | 242 (74%) | 281 (54%) | OR = 2.39 (1.77–3.23), $p < 0.001$† |
| Surfactant therapy, n (%) | 609 (72%) | 260 (80%) | 349 (68%) | OR = 1.87 (1.35–2.59), $p < 0.001$† |
| Mean airway pressure, mmHg (% of invasively ventilated infants with airway pressure available) | 8 | 8 (56%) | 8 (41%) | 0.2 mmHg (-0.37–0.78), $p = 0.49$* |

For 52% and 38% of the infants, inotropes were introduced within 10 hours of birth in Group 1 and 2, respectively. Almost all subjects had some form of airway support (99.6%). Small for gestational age (SGA), patent ductus arteriosus (PDA), intraventricular haemorrhage (IVH), odds ratio (OR), confidence interval (CI).

predicting whether infants needed surgical ligation after pharmacological interventions was 0.79. The area under the ROC curve for predicting the occurrence of IVH was 0.75. Additionally, we did a post hoc ROC analysis to compare the mean MAP value 4–10 hours divided with the mean MAP value 18–24 hours after birth (MAP ratio) to the rate of failed pharmacological interventions and IVH occurrence. MAP ratio poorly discriminated infants who needed surgical ligation after pharmacological interventions (AUC = 0.57). The discriminatory effect was marginally better for predicting the occurrence of IVH (AUC = 0.61). However, the mean MAP ratio was beneficial as an additional regressor for predicting the need for surgical PDA ligation and the occurrence of IVH. Our analysis did not suggest that using a larger mean MAP ratio would have resulted in better discrimination for failed pharmacological PDA interventions (optimal MAP ratio 0.99) or the occurrence of IVH (optimal MAP ratio 1.04).

## Discussion

Using a data-driven approach on an extensive data set of blood pressure measurements, we identified two unique blood pressure patterns during VLBW infants' transition to extrauterine life. We were able to operationalise the data-driven approach and identify a grouping principle

**Table 2. The multivariable logistic regression analysis results for variables predicting interventions of a patent ductus arteriosus.**

| | Pharmacological and surgical treatment of a PDA | | | | | | | |
| --- | --- | --- | --- | --- | --- | --- | --- | --- |
| | No treatment (n = 523) vs pharmacological treatment (n = 265) | | | | Pharmacological treatment vs pharmacological and surgical treatment (n = 83) | | | |
| | Odds ratio | nyes (%) / nno (%) | 95 CI for odds ratio | p-value | Odds ratio | nyes (%) / nno (%) | 95 CI for odds ratio | p-value |
| Gestational age (< 28 weeks, yes/no) | 1.56 | 147 (43) / 118 (24) | (1.03–2.38) | p = 0.037 | 5.29 | 70 (20) / 13 (3) | (2.37–11.8) | p < 0.001 |
| SGA (yes/no) | 0.64 | 40 (19) / 225 (36) | (0.41–1) | p = 0.052 | 0.44 | 7 (3) / 76 (12) | (0.16–1.22) | p = 0.114 |
| RDS (yes/no) | 1.56 | 174 (41) / 91 (21) | (1.07–2.27) | p = 0.02 | 1.22 | 56 (13) / 27 (6) | (0.63–2.37) | p = 0.557 |
| Sepsis (yes/no) | 1.16 | 60 (40) / 205 (29) | (0.77–1.75) | p = 0.475 | 0.83 | 19 (13) / 64 (9) | (0.4–1.71) | p = 0.615 |
| Antenatal corticosteroids (yes/no) | 0.98 | 251 (31) / 14 (39) | (0.46–2.09) | p = 0.951 | 3.82 | 80 (10) / 3 (8) | (0.93–15.7) | p = 0.063 |
| Fluid > 120 ml/kg (yes/no) | 1 | 152 (37) / 113 (26) | (0.7–1.44) | p = 0.985 | 1.99 | 58 (14) / 25 (6) | (1.01–3.93) | p = 0.047 |
| MAP18-24 < MAP4–10 hours after birth (Group 1, yes/no) | 0.92 | 109 (33) / 156 (30) | (0.65–1.3) | p = 0.639 | 2.1 | 44 (13) / 39 (8) | (1.14–3.89) | p = 0.018 |
| MAP < 33 mmHg 13–16 hours after birth (yes/no) | 1.26 | 75 (46) / 190 (28) | (0.8–1.97) | p = 0.316 | 1.25 | 33 (20) / 50 (7) | (0.63–2.46) | p = 0.52 |
| Inotrope (yes/no) | 2.35 | 178 (43) / 87 (20) | (1.63–3.37) | p < 0.001 | 0.55 | 58 (14) / 25 (6) | (0.26–1.15) | p = 0.11 |
| Invasive ventilation (yes/no) | 0.76 | 202 (39) / 63 (20) | (0.45–1.29) | p = 0.309 | 2.02 | 78 (15) / 5 (2) | (0.6–6.8) | p = 0.258 |
| Surfactant administration (yes/no) | 3.46 | 240 (39) / 25 (11) | (1.77–6.77) | p < 0.001 | 1.72 | 82 (13) / 1 (0) | (0.16–18.39) | p = 0.652 |

The table shows the odds ratio, confidence intervals and p-values for each variable. The table also shows in columns 3 and 7 the number and percentage of infants who had pharmacological or surgical PDA treatment according to the binary definition provided by each grouping variable. Patent ductus arteriosus (PDA), small for gestational age (SGA), respiratory distress syndrome (RDS), confidence interval (CI), mean arterial blood pressure (MAP).

**Table 3. The multivariable logistic regression analysis results for variables predicting the presence of intraventricular haemorrhage of any grade.**

| | IVH of any grade | | | |
| --- | --- | --- | --- | --- |
| | No IVH (n = 690) vs IVH of any grade (n = 142), missing information (n = 12) | | | |
| | Odds ratio | nyes (%) / nno (%) | 95 CI for odds ratio | p-value |
| Gestational age (< 28 weeks, yes/no) | 2.67 | 98 (28) / 44 (9) | (1.59–4.48) | p < 0.001 |
| SGA (yes/no) | 0.37 | 14 (6) / 128 (20) | (0.2–0.69) | p = 0.002 |
| RDS (yes/no) | 1.57 | 95 (23) / 47 (11) | (1.01–2.46) | p = 0.046 |
| Sepsis (yes/no) | 0.74 | 25 (17) / 117 (17) | (0.45–1.23) | p = 0.241 |
| Antenatal corticosteroids (yes/no) | 1.08 | 134 (17) / 8 (22) | (0.46–2.54) | p = 0.866 |
| Fluid > 120 ml/kg (yes/no) | 1.79 | 98 (24) / 44 (10) | (1.14–2.79) | p = 0.011 |
| MAP18-24 < MAP4–10 hours after birth (Group 1, yes/no) | 1.83 | 78 (24) / 64 (12) | (1.23–2.72) | p = 0.003 |
| MAP < 33 mmHg 13–16 hours after birth (yes/no) | 1.04 | 46 (28) / 96 (14) | (0.65–1.67) | p = 0.875 |
| Inotrope (yes/no) | 1.01 | 92 (22) / 50 (12) | (0.64–1.57) | p = 0.98 |
| Invasive ventilation (yes/no) | 0.7 | 116 (22) / 26 (8) | (0.35–1.4) | p = 0.308 |
| Surfactant administration (yes/no) | 2.07 | 131 (22) / 11 (5) | (0.84–5.1) | p = 0.116 |

The table shows the odds ratio, confidence intervals and p-values for each variable. The table also shows in column 3 the number and percentage of infants who had IVH according to the binary definition provided by each grouping variable. Intraventricular haemorrhage (IVH), small for gestational age (SGA), respiratory distress syndrome (RDS), confidence interval (CI), mean arterial blood pressure (MAP).

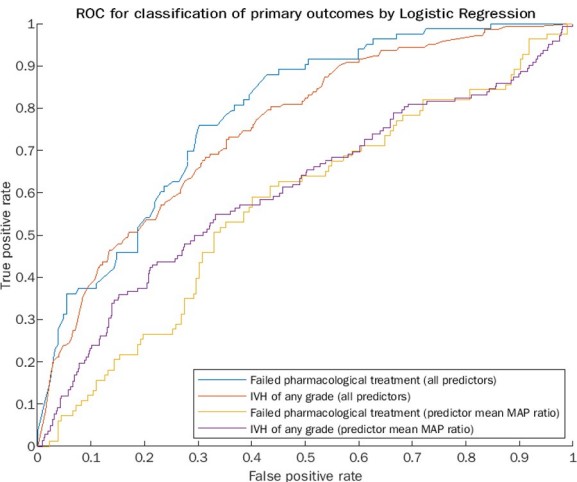

**Fig 3. Receiving operating characteristics analysis of the logistic regression models.** Receiving operating characteristics (ROC) analysis of the logistic regression models used for predicting the occurrence of intraventricular haemorrhage (IVH) and need for surgical patent ductus arteriosus (PDA) ligation after failed pharmacological interventions. Additionally, the figure displays a ROC analysis of the mean arterially recorded blood pressure (MAP) value 4–10 hours after birth divided with the mean MAP value 18–24 hours after birth to the rate of failed pharmacological interventions and occurrence IVH.

applicable to clinical work. Here we show that, in our cohort of VLBW infants, the infants with a lower mean MAP at 18–24 hours than at 4–10 hours after birth were at risk of a failed pharmacological PDA intervention and development of IVH.

This study adds to the existing knowledge of normal and pathological blood pressure changes during VLBW infants' adaption to extrauterine life. We showed that grouping infants according to blood pressure trends rather than absolute values helps predict whether a pharmacological PDA intervention will fail and whether the infants are susceptible to IVH. Although we used continuous blood pressure measurements to classify the infants into the two groups, classifying the infants based on hourly blood pressure measurements was also reasonably successful.

Previous studies have shown that a PDA is associated with the occurrence of RDS [1, 2, 4, 5], lower birthweight [1, 2, 4], lower gestational age [2, 4] and sepsis [4]. RDS and the severity of the lung disease [6] might be the most predictive factors for a PDA [4]. Our study repeated the finding that a shorter duration of pregnancy and the presence of RDS was associated with a pharmacological PDA intervention [9]. Factors previously associated with failed indomethacin treatment are low gestational age, lack of exposure to antenatal steroids, and RDS [31]. Here we repeated the finding that a shorter pregnancy duration was associated with an unsuccessful pharmacological PDA treatment.

Risk factors of IVH are complex and include RDS [32], a surgical PDA intervention [6], sepsis [33], hypotension [34–36], fluctuations in cerebral blood flow [37], and a low gestational age [36]. Antenatal corticosteroid administration protects from IVH [3]. We repeated the finding that RDS, high fluid intake [38] and low gestational age are risk factors for IVH. We failed to show an effect of antenatal corticosteroid treatment on developing a PDA and the occurrence of IVH. The reason is presumably the high proportion of mothers who received antenatal steroid treatment in our data (96%). Postnatal prophylactic steroids were not used in the patient cohort.

Hypotension treatment is a clinical challenge in VLBW infants, and optimal blood pressure ranges are debated [39]. Considering that infants with normal blood pressure often receive inotropes [40], restricting inotropes when treating premature infants could be beneficial [41].

However, it is worth noting that infants with hypotension have worse outcomes than infants with normal blood pressure [42]. Additionally, some cohort studies suggest that infants treated with either fluids, corticosteroids, or inotropes for hypotension have better short-term outcomes than infants with untreated hypotension [43]. Thus, recommending permissive hypotension requires randomised trials focusing on whether hypotension treatment improves the outcome of preterm infants [14]. However, fluid boluses for hypotensive infants during the first days of life may be associated with developing a PDA [44] and IVH [38]. Our finding that high fluid intake is associated with unsuccessful pharmacological PDA treatment and development of IVH highlights the importance of avoiding excessive fluid administration from day one.

Restricted fluid intake may be associated with a decreased risk of developing a PDA [7, 8, 45]. ESPGHAN/ESPEN/ESPR guidelines on paediatric parenteral nutrition suggest an upper limit of 90 ml/kg of fluids for infants with a birthweight between 1000 and 1500 g, and 100 ml/kg for infants of less than 1000 g, during the first day of life [29]. However, the ESPGHAN/ESPEN/ESPR guidelines add that certain clinical conditions could increase the total fluid need to approximately 120 ml/kg [29]. We knew from previous studies that the most premature infants' fluid intake exceeded 150 ml/kg on day one [16]. Thus, we used the upper limit (120 ml/kg) suggested by the ESPGHAN/ESPEN/ESPR guidelines to indicate excessive fluid administration during the first day of life. However, as fluid amounts of more than 146 ml/kg during the first day of life are associated with an increased risk of a PDA [8], using an even higher limit to indicate excessive fluid administration during the first day of life could have been justifiable.

There are numerous limitations to this study. One of this study's methodological weaknesses is that our primary endpoint (PDA treatment) was an intervention rather than a clinically standardised measurement. To address this methodological weakness, we included IVH as a primary endpoint. According to our NICUs practice, a PDA diagnosis requires focused cardiac ultrasound. Furthermore, pharmacological intervention is the primary approach for a PDA. Infants with a decreasing blood pressure trend did not receive a pharmacological PDA intervention more often than infants with rising blood pressure. Thus, our result is unlikely to reflect clinicians relying solely on a decreasing MAP trend to diagnose a clinically relevant PDA. It is worth noting that while early PDA treatment was used rather than expectant management during the study period, our NICU has shifted towards a more conservative approach when treating a PDA.

The exact time-point of IVH occurrence was not available in our data. According to our NICUs protocol, the first brain ultrasonography examination took place one day after birth. Thus, the predictor variables of this study, excluding sepsis, occurred before the first ultrasonography of the brain. However, to study whether the changes in blood pressure preceded or followed IVH, continuous IVH monitoring would have been necessary.

Regarding the reliability of the regressors, it is worth noting that more infants were on invasive ventilation than for whom we had records of mean airway pressure. However, only the recorded mean airway pressure indicated when the invasive ventilation was initiated and ended. Thus, the duration of invasive ventilation is not reliably reflected in our invasive ventilation regressor. Furthermore, part of the drugs and fluids administered in the delivery room may be unaccounted. On the one hand, the timestamps in the electronic patient system indicate the administration of fluids as soon as a few minutes after birth. On the other hand, the electronic patient information system suggested lower surfactant administration rates than the Finnish Medical Birth Register data on premature infants.

The "Golden Hour" principle assumes that patient care in the first hour of life is critical to the outcome [46, 47]. In our study sample, the blood pressure in both groups increased approximately seven hours after birth. This blood pressure increase is presumably part of

adaption after birth [10]. It is tempting to speculate that the decrease in blood pressure seen in Group 1 could have been avoided with appropriate treatment.

Low superior vena cava flow is associated with an increased risk of IVH [48]. Reasons for low vena cava flow includes high mean airway pressure and a ductal shunt [49]. The initial blood pressure rise seen in both groups could reflect the myocardium's ability to compensate for increased vascular resistance during adaption. In contrast, decreasing blood pressure could indicate failure of the adaptive mechanisms. A large ductal shunt could further diminish vena cava flow and accelerate the decrease in blood pressure. Low vena cava flow, decreasing cardiac output, increased vascular resistance, and failing adaptive mechanisms of the myocardium could result in a vicious circle explaining diminished cerebral blood flow and increased IVH occurrence in Group 1. In line with this hypothesis, more infants in Group 1 than in Group 2 were treated with invasive ventilation.

In summary, using a data-driven approach on 844 VLBW infants' blood pressure measurements, we identified two distinct blood pressure patterns during adaption to extrauterine life. Infants with a higher mean MAP at 4–10 hours than at 18–24 hours after birth had a higher occurrence of IVH and a failed pharmacological PDA intervention.

## Supporting information

**S1 Fig. Examining the clustering stability by repeating the k-means clustering analysis with bootstrap samples.** We repeated the k-means clustering analysis 4000 times with bootstrap samples to examine the clustering stability. The graph overlays the mean arterially recorded systolic, mean, and diastolic blood pressure measurements from each of the 4000 clustering analyses done with the bootstrap samples.
(TIF)

**S2 Fig. Arterially recorded blood pressure changes 24 hours after birth in infants with gestational ages 23–26, 27–30 and 31–34 weeks.** (A) The mean and the standard error for the systolic (blue), mean (red), and diastolic (yellow) arterially recorded blood pressure during the first 4–24 hours after birth. (B) The correlation matrix shows how k-means clustering divided the infants into two clusters based on how similar the infants' blood pressure time series were. (C) Arterially recorded blood pressure for infants in cluster 1. (D) Arterially recorded blood pressure for infants in cluster 2. Data are shown separately for infants with gestational age (GA) of (1.) 23–26 weeks, (2.) 27–30 weeks and (3.) 31–34 weeks. Infants with lower mean MAP at 18–24 hours than at 4–10 hours after birth and a GA of 23–26 weeks had a trend for an increased occurrence of IVH of grades III-IV (24% vs 14%, odds ratio = 1.97; CI 1–3.86; p = 0.059; Fischer's exact test). Infants with lower mean MAP at 18–24 hours than at 4–10 hours after birth and a gestational age of 27–30 weeks had an increased likelihood of needing surgical PDA interventions after pharmacological PDA interventions (9% vs 2%, odds ratio = 4.43; CI 1.7–11.56; p = 0.001; Fischer's exact test), and a trend for an increased occurrence of IVH of grades I–II (12% vs 7%, odds ratio = 1.8; CI 0.94–3.46; p = 0.09; Fischer's exact test) and IVH of grades III–IV (8% vs 4%, odds ratio = 2.27; CI 0.98–5.22; p = 0.055; Fischer's exact test). Infants with lower mean MAP at 18–24 hours than at 4–10 hours after birth and gestational age of 31–34 weeks had a trend for an increased occurrence of IVH of grades III-IV (4% vs 0%, odds ratio = NA; CI NA; p = 0.089; Fischer's exact test).
(TIF)

**S1 Table. The multinominal logistic regression analysis results for variables predicting surgical interventions of a patent ductus arteriosus.** The table shows the odds ratio, confidence intervals and p-values for each variable. The table also shows the number and percentage

of infants who had surgical PDA treatment according to the binary definition provided by each grouping variable. Patent ductus arteriosus (PDA), Small for gestational age (SGA), respiratory distress syndrome (RDS), confidence interval (CI), mean arterial blood pressure (MAP).
(DOCX)

**S2 Table. The multinominal logistic regression analysis results for variables predicting the risk of having no intraventricular haemorrhage (IVH) and having grade I–II or III–IV IVH.** The first column includes estimates for the odds of having no IVH vs having IVH of any grade. The second column includes estimates for the odds of having a grade I–II IVH versus a grade III–IV IVH, given that the infant had an IVH. The table shows the odds ratio, confidence intervals and p-values for each variable. The table also shows the number and percentage of infants who had IVH of grade I–II (third column) and IVH of grade III–IV (seventh column) according to the binary definition provided by each grouping variable. Small for gestational age (SGA), respiratory distress syndrome (RDS), confidence interval (CI), mean arterial blood pressure (MAP).
(DOCX)

**S1 File. Study material.** The data that support the findings of this study are contained in MAT files. The read_me.txt file describes contents of each mat file.
(ZIP)

## Acknowledgments

We thank Marita Suni, RN, for her help with this project.

## Author Contributions

**Conceptualization:** Robert Boldt, Reijo Sund, Markus Leskinen, Päivi Luukkainen, Sture Andersson.

**Data curation:** Robert Boldt, Pauliina M. Mäkelä, Lotta Immeli, Markus Leskinen.

**Formal analysis:** Robert Boldt, Pauliina M. Mäkelä, Lotta Immeli.

**Funding acquisition:** Sture Andersson.

**Investigation:** Robert Boldt, Pauliina M. Mäkelä, Lotta Immeli, Markus Leskinen, Päivi Luukkainen, Sture Andersson.

**Methodology:** Robert Boldt, Pauliina M. Mäkelä, Lotta Immeli, Reijo Sund, Markus Leskinen, Päivi Luukkainen.

**Project administration:** Robert Boldt, Sture Andersson.

**Resources:** Robert Boldt, Markus Leskinen, Sture Andersson.

**Software:** Robert Boldt.

**Supervision:** Reijo Sund, Markus Leskinen, Päivi Luukkainen, Sture Andersson.

**Validation:** Robert Boldt, Reijo Sund.

**Visualization:** Robert Boldt.

**Writing – original draft:** Robert Boldt, Pauliina M. Mäkelä, Lotta Immeli, Reijo Sund, Markus Leskinen, Päivi Luukkainen, Sture Andersson.

**Writing – review & editing:** Robert Boldt, Pauliina M. Mäkelä, Lotta Immeli, Reijo Sund, Markus Leskinen, Päivi Luukkainen, Sture Andersson.

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
