## [Decision Letter · Decision Letter 0]

11 Aug 2021

PONE-D-21-20256

Blood pressure changes during the first 4–24 hours of life in very low birthweight infants and susceptibility to a patent ductus arteriosus and intraventricular haemorrhage

PLOS ONE

Dear Dr. Boldt,

Thank you for submitting your manuscript to PLOS ONE. After careful consideration, we feel that it has merit but does not fully meet PLOS ONE’s publication criteria as it currently stands. Therefore, we invite you to submit a revised version of the manuscript that addresses the points raised during the review process.

Major changes requested are related to more detailed clinical status and outcomes in particular for discriminating low-grade to high-grade IVH. Quality data of all the regression models presented should be completed. In addition, ROC-curves would be of interest. Discussion section should be revised and improved, including a more comprehensive paragraph regarding the limitations of the study.

We look forward to receiving your revised manuscript.

Kind regards,

Olivier Baud, MD, PhD

Academic Editor

PLOS ONE

2. Please include captions for your Supporting Information files table at the end of your manuscript, and update any in-text citations to match accordingly. Please see our Supporting Information guidelines for more information: http://journals.plos.org/plosone/s/supporting-information.

Reviewers' comments:

Reviewer's Responses to Questions

**Comments to the Author**

1. Is the manuscript technically sound, and do the data support the conclusions?

Reviewer #1: Partly

Reviewer #2: Partly

2. Has the statistical analysis been performed appropriately and rigorously? 

Reviewer #1: I Don't Know

Reviewer #2: No

3. Have the authors made all data underlying the findings in their manuscript fully available?

Reviewer #1: Yes

Reviewer #2: No

4. Is the manuscript presented in an intelligible fashion and written in standard English?

Reviewer #1: Yes

Reviewer #2: Yes

5. Review Comments to the Author

Reviewer #1: Thank you for giving me the opportunity to review this study by Boldt R. et al. titled « Blood pressure changes during 1 the first 4–24 hours of life in very low birthweight infants and susceptibility to a patent ductus arteriosus and intraventricular haemorrhage ».

This retrospective cohort study involved 844 very low birth weight (VLBW) infants, ie with a birthweight of less than 1501 g or a gestational age at birth of less than 32+7 weeks admitted to the Helsinki University Children's Hospital during years 2005-2013.

The main objective was to explore mean blood pressure (MAP) changes during adaption, investigated through 600 time-points of MAP recorded for each infant in the NICU electronic patient information system. Blood pressure patterns were analyzed using k-means clustering, which allocated infants with similar blood pressure patterns to the same clusters. This method allowed to seperate two clusters :

*Group 1 (335 infants) who, after an initial rise between 4-10 hours after birth, had a declining blood pressure trend at 18-24 hours after birth.

*Group 2 (509 infants) who displayed a steadily rising trend throughout the study period (4-24 hours after birth).

The authors than assess whether this group assignments, and other variables, may predict the occurrence of intraventricular hemorrhage (IVH) and use of pharmacological or surgical therapy for a patent ductus arteriosus (PDA) before 42 weeks of gestational age.

For that purpose, the authors used data collected in the Finnish Medical Birth Register data on premature infants. They found that infants in Group 1 were more likely to undergo surgical ligation of a PDA (odds ratio = 2.06; CI 1.14–3.75; p = 0.017) and to suffer from IVH (odds ratio = 1.8; CI 1.22–2.66; p =0.003) when compared with infants in Group 2.

This study is quite interesting. Some consistency may be found with the observation of Kluckow and Evans, who found in prospective serial studies that low systemic blood flow (assessed by the means of the superior vena cava flow) occurs in about 35% of babies born before 30 weeks, with a nadir within the first 12 h after birth followed by a recovery of flow to normal values by 24 to 48 h. This low systemic blood flow was significantly associated with lower gestational age, higher mean airway pressures on the ventilator, larger ductal shunts, as with a range of adverse outcomes, particularly IVH [Kluckow M, Evans NJ. Low superior vena cava flow and intraventricular haemorrhage in preterm infants. Arch Dis Child 2000;82:F188-94].

The limitations of this study are numerous, as it involved a single center and a limited number of patients, the connection between different database, and above all it did not include any echocardiographic study in the first 24 hours of life making it possible to develop a well-constructed physiopathological hypothesis. In this regard, whatever the blood pressure patterns, gestational age and birthweight were significantly lower in Group 1, and it is hardly surprising that morbidity, particularly occurrence of PDA and IVH, was greater in this group.

I have some comments/suggestions:

- Could you please precise the death rate and the day of the first occurrence grades I-II and grades III-IV IVH (Table 1);

-Could you please precise the rates of invasive and non invasive ventilation in both groups, and if possible the maximum values for mean airway pressure in each group (Table 1);

- Could you please provide the results of the logistic regression model for no IVH or grade I-II IVH vs grade III-IV IVH;

- Could you please precise the quality of all the regression models presented, ie, adequacy test;

- Could you please estimate, for example using a ROC curve, the cut-off value of MAP ratio (eg, highest value 4-10 hours after birth /lowest value 18-24 hours after birth) predictive of IVH and PDA risk?

-In this perspective and using such kind of index, identifying a dose-effect relationship could suggest a causal relationship between a blood pressure pattern and the occurrence of these adverse events.

- The discussion is not informative, in particular it does not discuss the potential physiopathological link between the blood pressure patterns individualized by using k-means clustering and the morbidities observed in patients.

- The paragraph on study limitations should be much more detailed (please see the remark above)

- In addition, some key studies performed on blood pressure in premature infants and treatment of neonatal hypotension should also be cited and discussed (please see notably Batton B. et al J Pediatr 2009; Batton B. et al Pediatrics 2013; Faust K. et al Arch Dis Child F&N Ed 2015; Durrmeyer X. et al Arch Dis Child Fetal Neonatal Ed 2017; Dempsey EM. et al Arch Dis Child Fetal Neonatal Ed 2021.

Reviewer #2: I would like to thank the Editor and the Authors for the opportunity to review the manuscript entitled:Blood pressure changes during the first 4-24 hours of life in VLBW infants and susceptibility to a PDA and IVH.

The study focused on a problem of high relevance: the potential association of blood pressure variability during the first 24h of life on the PDA and the presence of IHV in premature infants. Authors found more PDA ligation and more IVH in group 1 (mean MAP higher at 4-10h than 18-24h) compared to group 2 (lower mean MAP at 4-10h than 18-24h).

I think your work is very interesting and at the same time original in the sense of being able to establish two groups of trends in blood pressure in the first 24h.

However, I have some comments and questions about possible confounding factors.

I think the title is misleading, perhaps it would be clearer to say: blood pressure changes or variability during the first 24 hours of life and the association with persistence of PDA and IVH.

Regarding the criteria for evaluation and diagnosis of ductus arteriosus. Sometimes you speak of clinically significant and others simply significant, could you clarify in more detail what you mean by significant? do you refer to clinical or echographic criteria? unfortunately this information is not reflected in your work.

You mention the surgical or pharmacological treatment of the ductus without mentioning the criteria for the indication of surgery or pharmacological treatment.

Regarding fluid intake, you mention that you have quantified fluid intake within the first 24h, have you included possible bolus in the delivery room or red blood cell transfusions in the first 24h?

Regarding the administration of vasoactive drugs, table 1 clearly shows that the first group received more vasoactive support than the second group. What is your protocol for introducing vasoactive support and at when do you consider hypotension? Could you specify in what period of time (4-10h v/s 18-24h) the vasoactive support was introduced or add that the period of amine administration was the same in both groups?

Why have vasoactive support not been included in the multivariable logistic regression?

In your article you describe that there was a blood pressure peak at 7h of life, have you seen in your cohort if there are patients of low gestational age who were able to exceed the 99th percentile of systolic, diastolic or mean pressure?

Regarding the intraventricular hemorrhages in table 1 show that group 1 has had more IVH than group 2, have you done a subgroup analysis to see if this difference is maintained independently of the early gestational age? The same considerations apply to the non response of pharmacological treatment of PDA.

It may be interesting to see if in different gestational age groups, for example between 23-26 GA, 27-30 GA and 31-34 GA, the difference in terms of the treatment of PDA and the presence of HIV is the same.

Regarding the K means clustering test, have you been able to see if the two patterns found (group 1 and 2) are repeated at different types of gestational age, i.e., between 23-26GA the K means test gives the same result as if you have a group between 27-30 GA and between 31-34 GA?

Another point that I think is important to point out is regarding the lack of information on the need for mechanical ventilation and surfactant administration considering that both could be associated with an increased risk of IVH.

In the abstract it seems to me that lines 37 and 38 say the same thing as lines 35 and 36.

I recommend to add the missing information and a subgroup analysis in order to give more strength to the differences you have obtained.

6. PLOS authors have the option to publish the peer review history of their article (what does this mean?). If published, this will include your full peer review and any attached files.

Reviewer #1: No

Reviewer #2: **Yes: **Francisca Barcos

---

## [Author Response · Author response to Decision Letter 0]

2 Oct 2021

PONE-D-21-20256

Blood pressure changes during the first 4–24 hours of life in very low birthweight infants and susceptibility to a patent ductus arteriosus and intraventricular haemorrhage

PLOS ONE

Dear Dr. Boldt,

Thank you for submitting your manuscript to PLOS ONE. After careful consideration, we feel that it has merit but does not fully meet PLOS ONE’s publication criteria as it currently stands. Therefore, we invite you to submit a revised version of the manuscript that addresses the points raised during the review process.

Major changes requested are related to more detailed clinical status and outcomes in particular for discriminating low-grade to high-grade IVH. Quality data of all the regression models presented should be completed. In addition, ROC-curves would be of interest. Discussion section should be revised and improved, including a more comprehensive paragraph regarding the limitations of the study.

A rebuttal letter that responds to each point raised by the academic editor and reviewer(s). You should upload this letter as a separate file labeled ‘Response to Reviewers’.

A marked-up copy of your manuscript that highlights changes made to the original version. You should upload this as a separate file labeled ‘Revised Manuscript with Track Changes’.

An unmarked version of your revised paper without tracked changes. You should upload this as a separate file labeled ‘Manuscript’.

We look forward to receiving your revised manuscript.

Kind regards,

Olivier Baud, MD, PhD

Academic Editor

PLOS ONE

Dear Dr Baud

We are grateful for the feedback and the possibility to revise our manuscript. As suggested, we now discriminate low-grade and high-grade IVH in the regression analysis. We also assess the model adequacy by presenting ROC curves for the logistic regression models. As suggested, we also present ROC curves addressing the relationship between blood pressure and surgical ductus interventions, and between blood pressure and intraventricular haemorrhage. We revised the discussion section by proposing a model explaining the blood pressure changes we describe. Furthermore, we improved the paragraph regarding the limitations of the study. 

Sincerely

Robert Boldt

Reviewers’ comments:

Reviewer’s Responses to Questions

Comments to the Author

1. Is the manuscript technically sound, and do the data support the conclusions?

Reviewer #1: Partly

Reviewer #2: Partly

2. Has the statistical analysis been performed appropriately and rigorously? 

Reviewer #1: I Don’t Know

Reviewer #2: No

3. Have the authors made all data underlying the findings in their manuscript fully available?

Reviewer #1: Yes

Reviewer #2: No

4. Is the manuscript presented in an intelligible fashion and written in standard English?

Reviewer #1: Yes

Reviewer #2: Yes

5. Review Comments to the Author

Reviewer #1: Thank you for giving me the opportunity to review this study by Boldt R. et al. titled « Blood pressure changes during 1 the first 4–24 hours of life in very low birthweight infants and susceptibility to a patent ductus arteriosus and intraventricular haemorrhage ».

This retrospective cohort study involved 844 very low birth weight (VLBW) infants, ie with a birthweight of less than 1501 g or a gestational age at birth of less than 32+7 weeks admitted to the Helsinki University Children’s Hospital during years 2005-2013.

The main objective was to explore mean blood pressure (MAP) changes during adaption, investigated through 600 time-points of MAP recorded for each infant in the NICU electronic patient information system. Blood pressure patterns were analyzed using k-means clustering, which allocated infants with similar blood pressure patterns to the same clusters. This method allowed to seperate two clusters :

*Group 1 (335 infants) who, after an initial rise between 4-10 hours after birth, had a declining blood pressure trend at 18-24 hours after birth.

*Group 2 (509 infants) who displayed a steadily rising trend throughout the study period (4-24 hours after birth).

The authors than assess whether this group assignments, and other variables, may predict the occurrence of intraventricular hemorrhage (IVH) and use of pharmacological or surgical therapy for a patent ductus arteriosus (PDA) before 42 weeks of gestational age.

For that purpose, the authors used data collected in the Finnish Medical Birth Register data on premature infants. They found that infants in Group 1 were more likely to undergo surgical ligation of a PDA (odds ratio = 2.06; CI 1.14–3.75; p = 0.017) and to suffer from IVH (odds ratio = 1.8; CI 1.22–2.66; p =0.003) when compared with infants in Group 2.

This study is quite interesting. Some consistency may be found with the observation of Kluckow and Evans, who found in prospective serial studies that low systemic blood flow (assessed by the means of the superior vena cava flow) occurs in about 35% of babies born before 30 weeks, with a nadir within the first 12 h after birth followed by a recovery of flow to normal values by 24 to 48 h. This low systemic blood flow was significantly associated with lower gestational age, higher mean airway pressures on the ventilator, larger ductal shunts, as with a range of adverse outcomes, particularly IVH [Kluckow M, Evans NJ. Low superior vena cava flow and intraventricular haemorrhage in preterm infants. Arch Dis Child 2000;82:F188-94].

The limitations of this study are numerous, as it involved a single center and a limited number of patients, the connection between different database, and above all it did not include any echocardiographic study in the first 24 hours of life making it possible to develop a well-constructed physiopathological hypothesis. In this regard, whatever the blood pressure patterns, gestational age and birthweight were significantly lower in Group 1, and it is hardly surprising that morbidity, particularly occurrence of PDA and IVH, was greater in this group.

I have some comments/suggestions:

- Could you please precise the death rate and the day of the first occurrence grades I-II and grades III-IV IVH (Table 1);

As suggested, we now state on page 8 rows 149-153. “In addition to the eight infants discarded from the study as they deceased before 24 hours of age, 61 infants died before 42 weeks of gestational age. Of these 61 infants, 29 died during the first week, 14 during the second week, three during the third week, three during the fourth week and 12 after the fourth week.”

Furthermore, we included the following on page 19, rows 382-386: “The exact time-point of IVH occurrence was not available in our data. According to our NICUs protocol, the first brain ultrasonography examination took place one day after birth. Thus, the predictor variables of this study, excluding sepsis, occurred before the first ultrasonography of the brain. However, to study whether the changes in blood pressure preceded or followed IVH, continuous IVH monitoring would have been necessary.” 

-Could you please precise the rates of invasive and non invasive ventilation in both groups, and if possible the maximum values for mean airway pressure in each group (Table 1);

As suggested, we now include the rates of invasive ventilation, surfactant administration and maximum airway pressure, when available, in table 1. Furthermore, we added the following to the footnote of table 1: “Almost all subjects had some form of airway support (99.6 %)”.

- Could you please provide the results of the logistic regression model for no IVH or grade I-II IVH vs grade III-IV IVH;

Thank you for this suggestion. We now included the suggested analysis in supplementary results Table 2.

- Could you please precise the quality of all the regression models presented, ie, adequacy test;

- Could you please estimate, for example using a ROC curve, the cut-off value of MAP ratio (eg, highest value 4-10 hours after birth /lowest value 18-24 hours after birth) predictive of IVH and PDA risk?

-In this perspective and using such kind of index, identifying a dose-effect relationship could suggest a causal relationship between a blood pressure pattern and the occurrence of these adverse events.

Thank you for this suggestion. We added the following text to page 14-15 rows 282-295 along with a new figure (Figure 3): “We did a post hoc receiving operating characteristics (ROC) analysis as a test of model adequacy of the logistic regression models (Figure 3) (30). The area under the ROC curve for predicting whether infants needed surgical ligation after pharmacological interventions was 0.79. The area under the ROC curve for predicting the occurrence of IVH was 0.75. Additionally, we did a post hoc ROC analysis to compare the mean MAP value 4-10 hours divided with the mean MAP value 18-24 hours after birth (MAP ratio) to the rate of failed pharmacological interventions and IVH occurrence. MAP ratio poorly discriminated infants who needed surgical ligation after pharmacological interventions (AUC = 0.57). The discriminatory effect was marginally better for predicting the occurrence of IVH (AUC = 0.61). However, the mean MAP ratio was beneficial as an additional regressor for predicting the need for surgical PDA ligation and the occurrence of IVH. Our analysis did not suggest that using a larger mean MAP ratio would have resulted in better discrimination for failed pharmacological PDA interventions (optimal MAP ratio 0.99) or the occurrence of IVH (optimal MAP ratio 1.04).”

- The discussion is not informative, in particular it does not discuss the potential physiopathological link between the blood pressure patterns individualized by using k-means clustering and the morbidities observed in patients.

We appreciate this suggestion and now included the following on page 20, rows 402-411: “Low superior vena cava flow is associated with an increased risk of IVH (47). Reasons for low vena cava flow includes high mean airway pressure and a ductal shunt (48). The initial blood pressure rise seen in both groups could reflect the myocardium's ability to compensate for increased vascular resistance during adaption. In contrast, decreasing blood pressure could indicate failure of the adaptive mechanisms. A large ductal shunt could further diminish vena cava flow and accelerate the decrease in blood pressure. Low vena cava flow, decreasing cardiac output, increased vascular resistance, and failing adaptive mechanisms of the myocardium could result in a vicious circle explaining diminished cerebral blood flow and increased IVH occurrence in Group 1. In line with this hypothesis, more infants in Group 1 than in Group 2 were treated with invasive ventilation.”

- The paragraph on study limitations should be much more detailed (please see the remark above)

- In addition, some key studies performed on blood pressure in premature infants and treatment of neonatal hypotension should also be cited and discussed (please see notably Batton B. et al J Pediatr 2009; Batton B. et al Pediatrics 2013; Faust K. et al Arch Dis Child F&N Ed 2015; Durrmeyer X. et al Arch Dis Child Fetal Neonatal Ed 2017; Dempsey EM. et al Arch Dis Child Fetal Neonatal Ed 2021.

We appreciate the suggestion regarding the incorporation of the suggested reference, which despite their relevance, were not considered in the original version. Additionally, we now included a more detailed discussion on study limitations on page 19-20, rows 371-396: “There are numerous limitations to this study. One of this study's methodological weaknesses is that our primary endpoint (PDA treatment) was an intervention rather than a clinically standardised measurement. To address this methodological weakness, we included IVH as a primary endpoint. According to our NICUs practice, a PDA diagnosis requires focused cardiac ultrasound. Furthermore, pharmacological intervention is the primary approach for a PDA. Infants with a decreasing blood pressure trend did not receive a pharmacological PDA intervention more often than infants with rising blood pressure. Thus, our result is unlikely to reflect clinicians relying solely on a decreasing MAP trend to diagnose a clinically relevant PDA. It is worth noting that while early PDA treatment was used rather than expectant management, during the study period, our NICU has shifted towards a more conservative approach when treating a PDA. 

The exact time-point of IVH occurrence was not available in our data. According to our NICUs protocol, the first brain ultrasonography examination took place one day after birth. Thus, the predictor variables of this study, excluding sepsis, occurred before the first ultrasonography of the brain. However, to study whether the changes in blood pressure preceded or followed IVH, continuous IVH monitoring would have been necessary.

Regarding the reliability of the regressors, it is worth noting that more infants were on invasive ventilation than for whom we had records of mean airway pressure. However, only the recorded mean airway pressure indicated when the invasive ventilation was initiated and ended. Thus, the duration of invasive ventilation is not reliably reflected in our invasive ventilation regressor. Furthermore, part of the drugs and fluids administered in the delivery room may be unaccounted for. On the one hand, the timestamps in the electronic patient system sometimes indicate the administration of fluids as soon as a few minutes after birth. On the other hand, the electronic patient information system suggested lower surfactant administration rates than the Finnish Medical Birth Register data on premature infants.”

Reviewer #2: I would like to thank the Editor and the Authors for the opportunity to review the manuscript entitled:Blood pressure changes during the first 4-24 hours of life in VLBW infants and susceptibility to a PDA and IVH.

The study focused on a problem of high relevance: the potential association of blood pressure variability during the first 24h of life on the PDA and the presence of IHV in premature infants. Authors found more PDA ligation and more IVH in group 1 (mean MAP higher at 4-10h than 18-24h) compared to group 2 (lower mean MAP at 4-10h than 18-24h).

I think your work is very interesting and at the same time original in the sense of being able to establish two groups of trends in blood pressure in the first 24h.

However, I have some comments and questions about possible confounding factors.

I think the title is misleading, perhaps it would be clearer to say: blood pressure changes or variability during the first 24 hours of life and the association with persistence of PDA and IVH.

Thank you for this suggestion. We now changed the title to read: “Blood pressure changes during the first 24 hours of life and the association with the persistence of a patent ductus arteriosus and occurrence of intraventricular haemorrhage”

Regarding the criteria for evaluation and diagnosis of ductus arteriosus. Sometimes you speak of clinically significant and others simply significant, could you clarify in more detail what you mean by significant? do you refer to clinical or echographic criteria? unfortunately this information is not reflected in your work.

You mention the surgical or pharmacological treatment of the ductus without mentioning the criteria for the indication of surgery or pharmacological treatment.

Thank you for this suggestion. We now included the following on page 7, rows 134-138: “According to our NICUs practice, diagnosis of a significant PDA requires indications of left heart volume overload, pulmonary hyperperfusion and focused cardiac ultrasound showing a moderate or large PDA. Surgical intervention is indicated if a significant PDA persists despite pharmacological interventions or if pharmacological interventions are contraindicated.”

Regarding fluid intake, you mention that you have quantified fluid intake within the first 24h, have you included possible bolus in the delivery room or red blood cell transfusions in the first 24h?

We now included the following on page 20, rows 391-396: “…part of the drugs and fluids administered in the delivery room may be unaccounted for. On the one hand, the timestamps in the electronic patient system sometimes indicate the administration of fluids as soon as a few minutes after birth. On the other hand, the electronic patient information system suggested lower surfactant administration rates than the Finnish Medical Birth Register data on premature infants.”

Regarding the administration of vasoactive drugs, table 1 clearly shows that the first group received more vasoactive support than the second group. What is your protocol for introducing vasoactive support and at when do you consider hypotension? Could you specify in what period of time (4-10h v/s 18-24h) the vasoactive support was introduced or add that the period of amine administration was the same in both groups?

We now included the following on page 7, rows 131-133: “The primary vasoactive agent in our NICU is dopamine. Additional vasoactive agents are noradrenalin, adrenalin and dobutamine. The MAP threshold for initiating vasoactive support in our NICU typically equals the infant's gestational age in weeks.”

We now included the following in the footnote of Table 1: “For 52 % and 38 % of the infants, vasoactive support was introduced within 10 hours of birth in Group 1 and 2, respectively.”

Why have vasoactive support not been included in the multivariable logistic regression?

Thank you for this suggestion. We now included vasoactive support in the logistic regression models.

In your article you describe that there was a blood pressure peak at 7h of life, have you seen in your cohort if there are patients of low gestational age who were able to exceed the 99th percentile of systolic, diastolic or mean pressure?

We now included the following on page 11, rows 214-216: “The blood pressure level was closely related to gestational age; only two infants with a gestational age < 27 weeks had a mean MAP higher than the mean MAP for all infants with a gestational age over 31 weeks.”

Regarding the intraventricular hemorrhages in table 1 show that group 1 has had more IVH than group 2, have you done a subgroup analysis to see if this difference is maintained independently of the early gestational age? The same considerations apply to the non response of pharmacological treatment of PDA.

It may be interesting to see if in different gestational age groups, for example between 23-26 GA, 27-30 GA and 31-34 GA, the difference in terms of the treatment of PDA and the presence of HIV is the same.

Regarding the K means clustering test, have you been able to see if the two patterns found (group 1 and 2) are repeated at different types of gestational age, i.e., between 23-26GA the K means test gives the same result as if you have a group between 27-30 GA and between 31-34 GA?

Thank you for this suggestion. We now repeated the analysis of the main manuscript for the suggested gestational age groups. We now present subgroup analyses in the supplementary materials (Supplementary Figure 2). 

Another point that I think is important to point out is regarding the lack of information on the need for mechanical ventilation and surfactant administration considering that both could be associated with an increased risk of IVH.

We now included the rate of mechanical ventilation, surfactant administration and use of vasoactive agents in Table 1 and the main regression analysis.

In the abstract it seems to me that lines 37 and 38 say the same thing as lines 35 and 36.

We now revised the abstract: “Infants whose mean MAP is higher at 4–10 hours than at 18–24 hours after birth are more likely to undergo surgical ligation of a PDA (odds ratio = 2.06; CI 1.14–3.75; p = 0.017) and to suffer from IVH (odds ratio = 1.8; CI 1.22–2.66; p = 0.003).”

I recommend to add the missing information and a subgroup analysis in order to give more strength to the differences you have obtained.

We now included the subgroup analysis in the supplementary materials.

---

## [Decision Letter · Decision Letter 1]

25 Oct 2021

PONE-D-21-20256R1Blood pressure changes during the first 24 hours of life and the association with the persistence of a patent ductus arteriosus and occurrence of intraventricular haemorrhagePLOS ONE

Dear Dr. Boldt,

Thank you for submitting your manuscript to PLOS ONE. After careful consideration, we feel that it has merit but does not fully meet PLOS ONE’s publication criteria as it currently stands. Therefore, we invite you to submit a revised version of the manuscript that addresses the points raised during the review process.

Both reviewers highly appreciate your efforts to address their points raised during the previous round of review. I kindly ask you to respond to the outstanding comments raised by reviewer 2.

We look forward to receiving your revised manuscript.

Kind regards,

Harald Ehrhardt

Academic Editor

PLOS ONE

Journal Requirements:

Reviewers' comments:

Reviewer's Responses to Questions

**Comments to the Author**

1. If the authors have adequately addressed your comments raised in a previous round of review and you feel that this manuscript is now acceptable for publication, you may indicate that here to bypass the “Comments to the Author” section, enter your conflict of interest statement in the “Confidential to Editor” section, and submit your "Accept" recommendation.

Reviewer #1: All comments have been addressed

Reviewer #2: All comments have been addressed

2. Is the manuscript technically sound, and do the data support the conclusions?

Reviewer #1: Yes

Reviewer #2: Yes

3. Has the statistical analysis been performed appropriately and rigorously? 

Reviewer #1: Yes

Reviewer #2: Yes

4. Have the authors made all data underlying the findings in their manuscript fully available?

Reviewer #1: Yes

Reviewer #2: Yes

5. Is the manuscript presented in an intelligible fashion and written in standard English?

Reviewer #1: Yes

Reviewer #2: Yes

6. Review Comments to the Author

Reviewer #1: To my opinion, this second version of the manuscript is significantly improved compared to the first one. The authors have addressed a majority of my points. I have no more question and congratulate the authors for their study.

Best regards.

Reviewer #2: I would like to thank the Editor and the Authors for the opportunity to

re-review the manuscript entitled now: “Blood pressure

changes during the first 24 hours of life and the association with the persistence of a

patent ductus arteriosus and occurrence of intraventricular haemorrhage”.

Thank you very much for responding to all comments and questions and for performing the requested analysis.

I have some questions. have you used in your cohort postnatal prophylactic corticosteroids?

I have some suggestions for a better visibility of your data:

Would it be possible to convert table 2 into 2 tables, it seems to me that there is a lot of information in table 2. One in relation to PDA and the other with HIV?

From my point of view I think it would be better to say that group 1 is the group that has had the lower mean MAP at 18-24h instead of saying that group 1 has had the higher mean MAP at 4-10h. For me it’s much clearer and the message makes more sense, but it’s just a personal opinion. It's up to you if you want to change this.

7. PLOS authors have the option to publish the peer review history of their article (what does this mean?). If published, this will include your full peer review and any attached files.

Reviewer #1: No

Reviewer #2: **Yes: **Francisca Barcos-Munoz

---

## [Author Response · Author response to Decision Letter 1]

2 Nov 2021

PONE-D-21-20256R1

Blood pressure changes during the first 24 hours of life and the association with the persistence of a patent ductus arteriosus and occurrence of intraventricular haemorrhage

PLOS ONE

Dear Dr. Boldt,

Thank you for submitting your manuscript to PLOS ONE. After careful consideration, we feel that it has merit but does not fully meet PLOS ONE's publication criteria as it currently stands. Therefore, we invite you to submit a revised version of the manuscript that addresses the points raised during the review process.

Both reviewers highly appreciate your efforts to address their points raised during the previous round of review. I kindly ask you to respond to the outstanding comments raised by reviewer 2.

We look forward to receiving your revised manuscript.

Kind regards,

Harald Ehrhardt

Academic Editor

PLOS ONE

Dear Prof. Dr. med. Ehrhardt, 

Thank you for allowing us to revise our manuscript. We want to thank the editors and the reviewers for their valuable comments. Furthermore, we thank reviewer 1 for previous valuable comments and your appreciation of our study. 

Reviewer 2 

We are grateful for the valuable insights you provided us during the major and minor revisions. Below, we address each comment raised during the latter review process. 

Reviewer #2: I would like to thank the Editor and the Authors for the opportunity to

re-review the manuscript entitled now: "Blood pressure

changes during the first 24 hours of life and the association with the persistence of a

patent ductus arteriosus and occurrence of intraventricular haemorrhage".

Thank you very much for responding to all comments and questions and for performing the requested analysis.

I have some questions. have you used in your cohort postnatal prophylactic corticosteroids?

On page 19, lines 357-358 of the manuscript, we now state the following:

"Postnatal prophylactic steroids were not used in the patient cohort."

I have some suggestions for a better visibility of your data:

Would it be possible to convert table 2 into 2 tables, it seems to me that there is a lot of information in table 2. One in relation to PDA and the other with HIV?

Thank you for the suggestion. We now divided table 2 into two separate tables (Table 2 page 15 and Table 3 page 16).

From my point of view I think it would be better to say that group 1 is the group that has had the lower mean MAP at 18-24h instead of saying that group 1 has had the higher mean MAP at 4-10h. For me it's much clearer and the message makes more sense, but it's just a personal opinion. It's up to you if you want to change this.

To improve the reading experience, we decided to change the wording as proposed.

---

## [Decision Letter · Decision Letter 2]

9 Nov 2021

Blood pressure changes during the first 24 hours of life and the association with the persistence of a patent ductus arteriosus and occurrence of intraventricular haemorrhage

PONE-D-21-20256R2

Dear Dr. Boldt,

We’re pleased to inform you that your manuscript has been judged scientifically suitable for publication and will be formally accepted for publication once it meets all outstanding technical requirements.

Kind regards,

Harald Ehrhardt

Academic Editor

PLOS ONE

Additional Editor Comments (optional):

Reviewers' comments:

Reviewer's Responses to Questions

**Comments to the Author**

1. If the authors have adequately addressed your comments raised in a previous round of review and you feel that this manuscript is now acceptable for publication, you may indicate that here to bypass the “Comments to the Author” section, enter your conflict of interest statement in the “Confidential to Editor” section, and submit your "Accept" recommendation.

Reviewer #2: All comments have been addressed

2. Is the manuscript technically sound, and do the data support the conclusions?

Reviewer #2: Yes

3. Has the statistical analysis been performed appropriately and rigorously? 

Reviewer #2: Yes

4. Have the authors made all data underlying the findings in their manuscript fully available?

Reviewer #2: Yes

5. Is the manuscript presented in an intelligible fashion and written in standard English?

Reviewer #2: Yes

6. Review Comments to the Author

Reviewer #2: This third version of the manuscript is, in my opinion considerably better than the first one, easier to read and understand.. I have no more question and congratulate the authors for their study.

Cordial greetings!

7. PLOS authors have the option to publish the peer review history of their article (what does this mean?). If published, this will include your full peer review and any attached files.

Reviewer #2: **Yes: **Francisca Barcos-Munoz

---

## [Editor Report · Acceptance letter]

19 Nov 2021

PONE-D-21-20256R2 

Blood pressure changes during the first 24 hours of life and the association with the persistence of a patent ductus arteriosus and occurrence of intraventricular haemorrhage 

Dear Dr. Boldt:

I'm pleased to inform you that your manuscript has been deemed suitable for publication in PLOS ONE. Congratulations! Your manuscript is now with our production department. 

Kind regards, 

on behalf of

Prof. Harald Ehrhardt 

Academic Editor

PLOS ONE